# Assessing the Freshwater Quality of a Large-Scale Mining Watershed: The Need for Integrated Approaches

**Daniel Mercado-Garcia** [1,*], **Eveline Beeckman** [1], **Jana Van Butsel** [1], **Nilton Deza Arroyo** [2],
**Marco Sanchez Peña** [2], **Cécile Van Buggendhoudt** [1], **Nancy De Saeyer** [1,3],
**Marie Anne Eurie Forio** [1], **Karel A. C. De Schamphelaere** [3], **Guido Wyseure** [4]
**and Peter Goethals** [1]

1   Aquatic Ecology Research Unit (AECO), Department of Animal Sciences and Aquatic Ecology,
    Ghent University, Coupure Links 653, 9000 Ghent, Belgium
2   Facultad de Ciencias de la Salud, Universidad Nacional de Cajamarca, Av. Atahualpa 1050,
    Cajamarca 06003, Perú
3   Environmental Toxicology Research Unit (GhEnToxLab), Department of Animal Sciences and Aquatic
    Ecology, Ghent University, Coupure Links 653, 9000 Ghent, Belgium
4   Division of Soil and Water Management, Department of Earth and Environmental Sciences,
    KU Leuven, Celestijnenlaan 200E, 3001 Leuven, Belgium
*   Correspondence: daniel.mercadogarcia@ugent.be; Tel.: +32-9-264-3895

**Abstract:** Water quality assessments provide essential information for protecting aquatic habitats and stakeholders downstream of mining sites. Moreover, mining companies must comply with environmental quality standards and include public participation in water quality monitoring (WQM) practices. However, overarching challenges beyond corporate environmental responsibility are the scientific soundness, political relevance and harmonization of WQM practices. In this study, a mountainous watershed supporting large-scale gold mining in the headwaters, besides urban and agricultural landuses at lower altitudes, is assessed in the dry season. Conventional physicochemical and biological (Biological Monitoring Water Party-Colombia index) freshwater quality parameters were evaluated, including hydromorphological and land-use characteristics. According to the indicators used, water quality deterioration by mining was absent, in contrast to the effects of urban economic activities, hydromorphological alterations and (less important) agricultural pollutants. We argue that mining impacts are hardly captured due to the limited ecological knowledge of high-mountain freshwaters, including uncharacterized mining-specific bioindicators, environmental baselines and groundwater processes, as well as ecotoxicological and microbial freshwater quality components. Lessons for overcoming scientific and operational challenges are drawn from joint efforts among governments, academia and green economy competitiveness. Facing a rapid development of extractive industries, interinstitutional and multidisciplinary collaborations are urgently needed to implement more integrated freshwater quality indicators of complex mining impacts.

**Keywords:** gold mining; urban impacts; agriculture; Andes mountains; benthic bioindicators

## 1. Introduction

Inland mining, in combination with urban growth and climate change, constitutes a major challenge for sustainable development, hindering water access to the most vulnerable stakeholders [1]. Thus, the monitoring and assessment of mining impacts is mandatory for protecting social-environmental assets [2–4]. Besides water balance programs by mining companies [5], freshwater quality maintenance

is vital for downstream users [4]. The assessment of mining impacts on freshwater quality is founded on the risk of releasing chemicals (e.g., ions, toxicants or metals) and ore exploitation debris in water bodies [6]. As obliged by environmental regulations, water quality monitoring (WQM) helps environmental managers to access information on pollutant sources and pathways in river systems [7]. However, mining sustainability is a "much more complex picture" and requires continuous technoscientific developments [5], suggesting that WQM practices require more integrated [7] and systemic [8,9] contributions.

*The Importance of Advancing Water Quality Monitoring (WQM) in Andean Mining Regions*

Despite the environmental risk of exploiting virgin ores, mining remains central to national economies specialized on metal exports [10]. For example, the abundance of rich ore deposits in Peru translates into a high percentage in the area of Andean watersheds claimed by mining concessions [11], and new mining projects are expected due to the strong economic dependence on mining [12]. Moreover, the Peruvian Andes are also known for hosting various ecoregions and natural resource hotspots [13], including areas with an adequate climate for housing, agriculture and tourism [14]. However, while the natural capital availability and political economy in Peru attract large mining investments, the assessment of mining impacts on watersheds encounters challenges at the institutional [11,15] and technical [10,16] levels. Water-related social conflicts, the polarized governance of land and water resources and the insufficient characterization of biophysical processes and land-use patterns are among the challenges for sustainable mining in Peru [15,16]. Such a complex social–ecological transition entails the robust generation, management and policy relevance of environmental monitoring data [17]. Therefore, advancing water quality assessments is becoming imperative for countries in which increasing urbanization, agriculture and mining threaten freshwater resources [18,19].

Researchers claim that human pressures and ecosystem dynamics in the Andes can be "radically different from preconceived ideas" [20], and that current knowledge of high-mountain freshwater ecology is insufficient to assess mining impacts [21]. Traditionally, WQM in mining contexts focuses on locations prone to receiving mining pollutants. Field and laboratory measurements are compared to well-defined quality standards for mining activities [6]. Such an approach provides information for remedial responses (e.g., pH or sediment control) deemed as "responsible mining" since environmental laws are being complied with. However, such a traditional WQM approach has not been conceived for the sustainability of freshwater ecosystems withstanding the multiple and complex impacts of mining. Lifecycle stages of mining (i.e., exploration, mine development, operation and closure) differ from each other in terms of the induced social–economic changes and in the types, severity and accumulation of environmental impacts [22]. Moreover, suboptimal environmental assessments can arise from the excessive reliance on quantitative methods while neglecting relevant metrics, or when disregarding important qualitative differences [23]. In the case of WQM, pollution can remain undetected either due to different spatiotemporal scales of occurrence, instrumental and methodological limitations, or due to attenuation by physical, chemical and biological processes. In the latter regard, benthic bioindicators are proven tools for capturing the effects of long-term environmental interactions as well as of abrupt water quality changes [24]. Physicochemical measurements in mining watersheds are thus complemented by identifying benthic macroinvertebrates or diatoms [6,25,26].

The recognition of watersheds as functional "living" entities (i.e., complex systems) implies that the dynamic nature of social–ecological systems influences the pressures and interactions on the overall system [7,9]. Moreover, freshwater quality variations are better represented by variations in the biotic–abiotic interplay of ecological components (i.e., ecological functioning) which are able to maintain freshwater ecosystems [27]. Overall, the WQM of mining watersheds requires an integrated approach using scientific, social and political assets. Pragmatically, monitoring data should facilitate the transparent communication and management of (post-)mining impacts, including water quality information further downstream of the mines [1]. However, budgetary and technical constraints, as well as the predominance of physicochemical water quality thresholds over ecological effects in

environmental norms [8], hinder WQM developments in this direction. In this regard, we assess the freshwater ecological quality of a mountainous watershed impacted by open-pit gold mining, agriculture and urbanization in Peru to identify challenges requiring broader—i.e., systemic—solutions aiding the soundness of practices and decisions around the WQM of mining watersheds.

## 2. Materials and Methods

### 2.1. Case Study

The Mashcon catchment (Figure 1) is the main hydrological system in Cajamarca province, covering 312 km$^2$ of the northern Peruvian Andes. The rough topography of the catchment makes the main watercourse drop from 3400 to 2600 m.a.s.l. (Figure 1b) over a distance of 30 km [28]. Above 3000 m.a.s.l., the Jalca ecosystem has transitioned to agricultural, plantation forestry and mining landuses [29]. At midstream altitudes, around 1.5% of the river Grande's water is captured to supply 70% of the city of Cajamarca's water [28]. Urban growth and rural displacement in Cajamarca coincide with the mining economic boom [12,14]. One of the largest gold mines in the world, Minera Yanacocha SRL (MYSRL), has operated at the headwaters since 1993. MYSRL has the permission to withdraw groundwater from open pits at a rate of 570 l/s, or 17.976 Mm$^3$ per year, securing the working environment by drying the pits [30]. Moreover, the river Grande is artificially recharged at the mining camp to compensate for the groundwater abstraction [16]. Non-mining stakeholders' concerns for mining impacts are related to decreasing phreatic levels [16] and risks of water pollution due to a mercury spill accident in the year 2000 [31]. Mining activities are thus at the center of Cajamarca's water resource controversy, although they comply with Peruvian regulations, generate fiscal revenues and provide community support near to the mines [15,32]. For more information, see Supplementary Material S1.

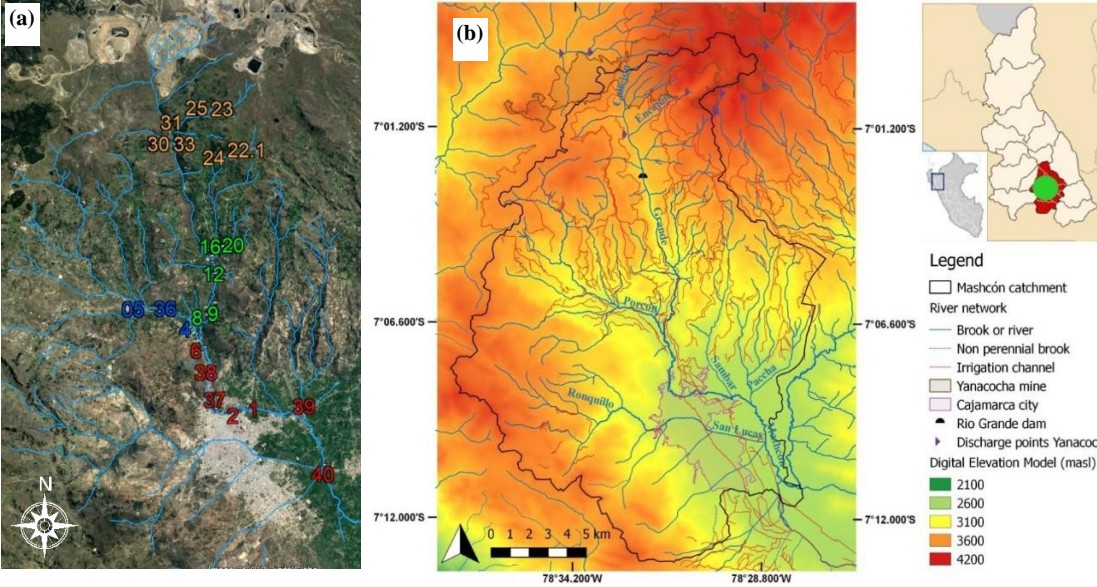

**Figure 1.** The Mashcon river basin. (**a**) Aerial photo showing the mining camp in the north (headwaters) and the city in the south (downstream), with sampling sites' labels coloured in red ('city'), blue ('Porcon'), green ('midstream') and orange ('mine'); (**b**) location and digital elevation model of the watershed.

### 2.2. Ecological Quality Assessment

A total of 40 sites were sampled in the Mashcon catchment in 2016 during the dry season. The driest period takes place between July and August in Cajamarca, thus allowing sample collection for a representative assessment of water quality in relation to adjacent land-uses [33]. Due to the expected altitudinal stratification of biophysical characteristics and human impacts (Table 1), monitoring sites

were divided into four groups or subsystems: (i) the mine (headwaters) and (ii) midstream, with 14 sites each; and (iii) Porcón and (iv) the city, with 4 and 8 sites, respectively. The Porcón is a tributary of the river Mashcon in which no metal mining occurs. The headwaters are in the Jalca, which is an Andean transition between the northern paramo and the puna further south. Below 3500 m.a.s.l., the local ecosystem is known as quechua. For more detailed information, see Supplementary Material S1.

**Table 1.** Criteria to define freshwater sampling groups in the Mashcon watershed.

| Subsystem | m.a.s.l. | Biophysical Characteristics | Anthropogenic Pressures on Rivers |
|---|---|---|---|
| Mine | 3270–3570 | More rainfall and UV radiation. Mainly herbaceous vegetation. Lower oxygen partial pressure. | Mining camp. Few houses and farms. Artificial headwaters recharge. Concrete channelling and bridges. |
| Midstream | 2800–2960 | Combination of shrubs, trees and herbaceous vegetation. Pristine hydromorphology with pool–riffle sequences. | Extensive agriculture. Greenhouses. Dirt road network. Scattered rural community infrastructures. Water capture plant for the city. |
| Porcon | 2780–2870 | Planted trees. Less riverbank vegetation. Less rainfall. Higher oxygen partial pressure. | Concrete floors, bridges and roads. Few farmlands. Riverbank stone extraction and granite factories. |
| City | 2660–2780 | Biophysical characteristics resemble the Porcón group, but the valley is much broader. | Most urbanized area. Anthropized riverbanks and open littering. Discharge of untreated wastewater. |

The physicochemical water quality was measured on-site using YSI® multiparameter probes with temperature, electric conductivity (EC), pH, turbidity, chlorophyll-a and dissolved oxygen (DO) detectors. Water samples were preserved for laboratory analyses using Hach-Lange® test kits for chemical oxygen demand (COD), nitrogen, phosphorus and organic ion concentrations. Sulphates were analysed using Hach® SulfaVer® 4 kits. Additionally, 50 mL of freshwater was filtered (0.45 μm), acidified to 1% $HNO_3$, and used for metal analyses of cadmium, barium, copper, chromium, iron, nickel, manganese, lead, zinc, cobalt and arsenic in a Thermo Fisher Scientific® iCAP 6000 ICP-OES®. Benthic bioindicators were collected by submerging a hand-net (frame size 20 × 30 cm; mesh size 500 μm) and applying the kick-sampling method over a stretch of 10–20 m. The simultaneous kicking and sweeping of the river substrate was done as evenly as possible at multiple micro-habitats for 5 min. Macroinvertebrates were thoroughly sorted, preserved in 75% ethanol, and identified at the family level. Geocoordinates and altitude were recorded using a Garmin eTrex® HC series GPS. Hydromorphology and landuse were registered using a standard field protocol (Supplementary Material S1, Table C1).

*2.3. Data Processing*

The longitudinal evolution of physicochemical variables in the main water course was plotted to detect changes in water quality, including pollutant peaks and linear patterns. Summary statistics were obtained after pooling the full dataset in 3 groups: (i) the main water course, consisting of the river Grande and river Mashcon; (ii) Porcon, a tributary of river Mashcon; and (ii) tributaries, consisting of all remaining streams of third order and below (i.e., tributaries of the river Grande). The distributions of physicochemical measurements among these three groups were compared using boxplots and summary statistics. Physicochemical measurements were likewise compared to the Peruvian environmental quality standards for streams intended for potable water production (D.S. N° 004-2017-MINAM; Law N° 28611) [34]. Two quality standards were considered: (i) the thresholds for freshwater streams requiring disinfection, and (ii) the thresholds for streams requiring advanced treatment. Physicochemical quality was also determined with the composite indices Prati and the Water Quality Index (WATQI) [35,36]. For this research, the Prati index was calculated based on five variables: DO saturation (%), pH, COD (mg/L), ammonium-N (mg N/l) and nitrate-N (mg N/l). Likewise, WATQI was based on DO (mg/L), EC (μS/cm), pH, total phosphorous (mg/L) and total nitrogen (mg/L). Biological

water quality was determined using the adapted index for benthic macroinvertebrates, BMWP-Col [37]. Spearman's rank correlation coefficients were calculated for all variables using R [38]. Data analysis graphs were generated in the R Studio software using ggplot2 [39], and the maps were produced in Quantum GIS version 2.18 and Google Earth®.

## 3. Results

This section describes physicochemical water quality variations first in order to then make a connection with both biotic and abiotic quality components (i.e., ecological quality changes in relation to anthropogenic pressures and natural variations of the watershed).

### 3.1. Varying Physicochemical Quality in the Main Water Course

Figure 2 shows the upstream-to-downstream physicochemical quality variation in the main water course, conforming to the river Grande coming from the mining camp (white background in Figure 2) and the river Mashcon flowing alongside urban areas (grey background in Figure 2). Values of pH remained around neutral on average despite a slight downstream increment (Figure 2c, blue line). DO increased in a dam at sampling site 2 (city) and showed its highest peak downstream of this dam (concentration of 10.8 mg/L, which equals 153% oxygen local saturation). The last sampling sites (i.e., the city outskirts) had plummeting DO concentrations (Figure 2c, green line) and organic pollutant concentrations far above the average values (Figure 2e–h). Smaller peaks of EC (Figure 2b), chlorophyll-a (Figure 2c), total nitrogen and nitrate (Figure 2b) were found in rural areas. Sulphate concentrations fluctuated at midstream sites, while mine and urban sites had lower concentrations and fewer fluctuations (Figure 2d).

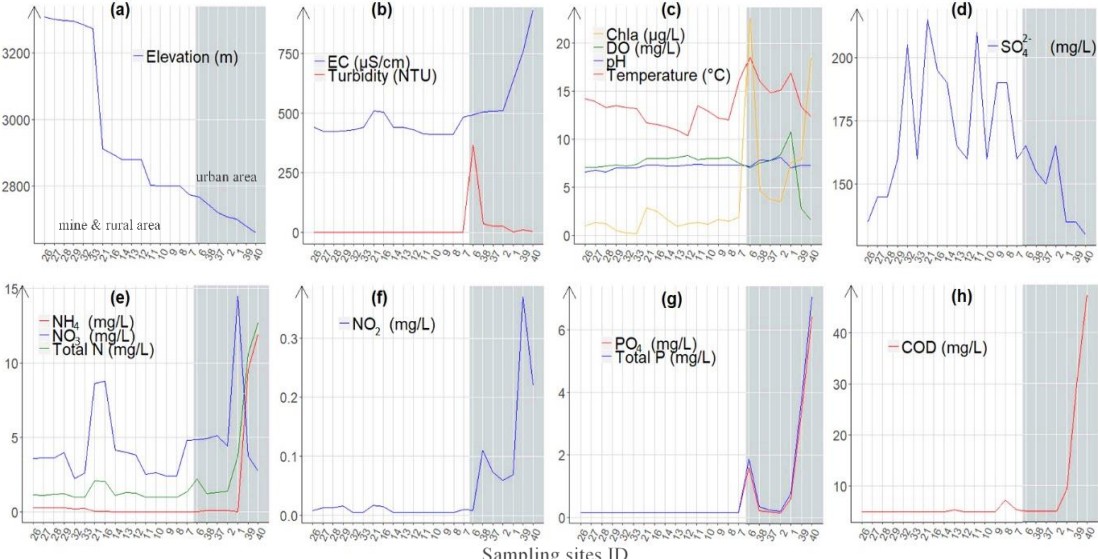

**Figure 2.** Physicochemical longitudinal gradient in the main water course conforming to the river Grande (white background = mine and midstream subsystems) and river Mashcon (grey background = city subsystem), showing a clear exacerbation of pollution at urban areas. (**a**): altitude. (**b**): electric conductivity and turbidity. (**c**): chlorophyll-a, dissolved oxygen, pH and temperature. (**d**): sulphates. (**e**): ammonia, nitrate and total nitrogen. (**f**): nitrite. (**g**): phosphates and total phosphorus. (**h**): chemical oxygen demand.

### 3.2. Water Quality Changes Near the Mine

The physicochemical quality in the upstream river Grande was stable (Figure 2), whereas the larger variabilities of sulphate, EC and pH measurements in the boxplots for the mine group (Figure A1) are due to the tributaries of the river Grande.

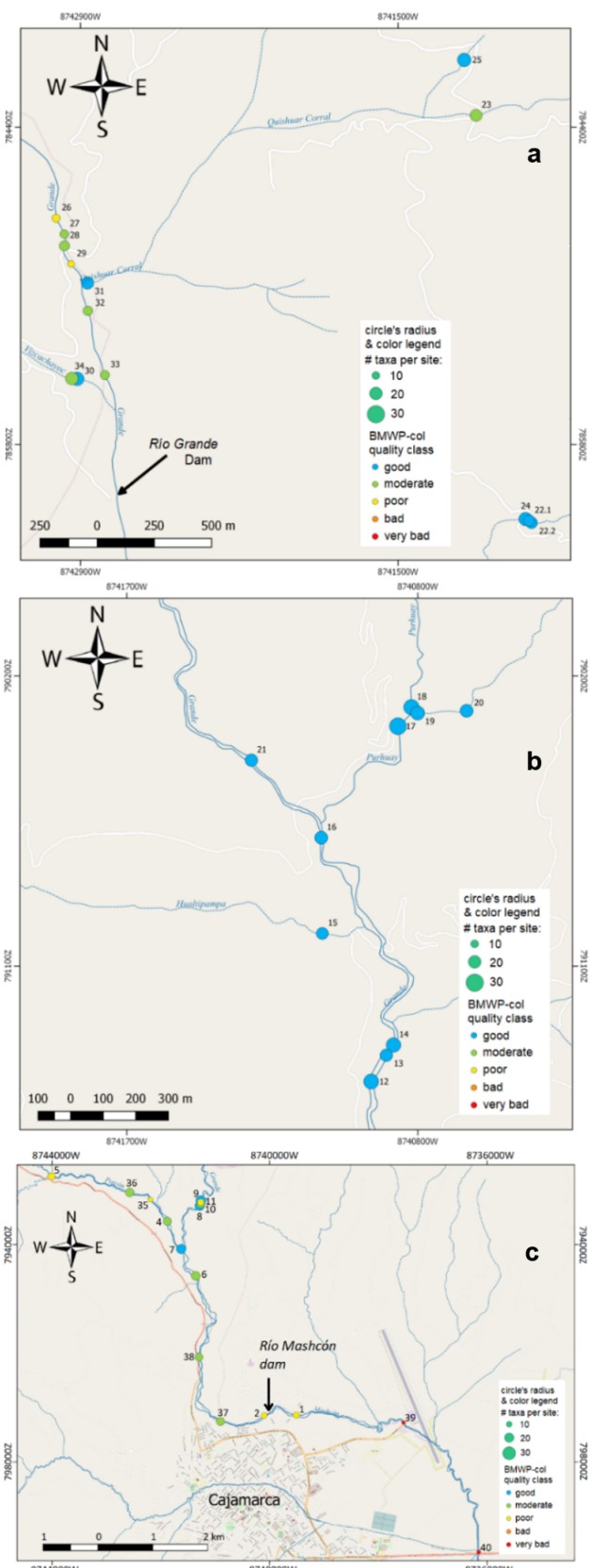

**Figure 3.** Index of biological quality, Biological Monitoring Water Party-Colombia, at the (**a**) mine subsystem, (**b**) midstream and (**c**) urban sections of the watershed. Sampling sites are identified with a number and a circle. The circle's radius is proportional to the number of taxa at each site and the color legend represents the BMWP-Col quality classes, with blue = good, green = moderate, yellow = poor, orange = bad, and red = very bad qualities.

The biological quality in the main watercourse (Figure 3a) was either poor or moderate at sites with channeled watercourses or walled riverbanks. Moreover, the mine subsystem had no signs of oxygen depletion, nor of increasing EC, multiple dissolved metal peaks (Table A1) or biological impairment (Figure 3a).

A slightly acidic pH and an iron pollution peak were found at site 25 (Table A1), although with a good biological quality (Figure 3a). Contrary to site 25, site 23 had a neutral pH and measurable iron of 0.006 mg/L, although with a moderate biological quality. Two high-altitude tributaries in the opposite mountain reach (sites 30 and 34, Table A1) showed pH values of 5.7 and 4.6. Furthermore, more alkalinity corresponded to an increase in urban pollution further downstream.

In addition, Acari insects were present in 60% of the monitoring sites but were remarkably more abundant near the San Jose mining waste centre (>110 Acari individuals, both at site 23 and site 25). Likewise, Grypoterygidae bioindicators in the mine group were only present in the main water course and in one tributary (site 31), while they were absent in the rest of the sampling points.

### 3.3. Water Quality Changes in Relation to Rural and Urban Pressures

Figures A1 and A2 show water quality deviations, often attributable to economic activities. Such deviations are represented by most of the outliers for each data subset (i.e., following the division presented in Table 1) considered for generating the boxplots, consisting from left to right in Figures A1 and A2 of the city points (first boxplot), midstream (second boxplot), mine (third boxplot) and Porcón (fourth boxplot).

Physicochemical impacts from ryegrass cultivation were found at the headwaters of the Liclipampa, a second-order tributary. A 12-fold increase in ammonium and a doubling nitrate concentration evince the effects of an irrigation outflow. Increments in EC, COD, chlorophyll-a, total nitrogen and sulphate were also detected in the latter context. Comparable nitrate increments were measured in a more voluminous stream—the fifth-order river Grande. However, impacts on the number of macroinvertebrate families were absent at sites where agricultural inputs influenced physicochemical quality (Figure 3b and bottom-right corner of Figure 3a). Likewise, we found that river Porcón's hydromorphology was severely impacted by economic activities, but few influences on physicochemical variables were detected (Table A1). The confluence of the rivers Porcón and Grande was less anthropized and showed good biological quality (Figure 3c, site 7) and thus healthier microhabitats than the upstream river Porcón. Unfortunately, 600 meters downstream, the biological quality was degraded again to a moderate class (Figure 3c, site 6) and turbidity showed its highest peak (Figure 2b) exactly where granite factories release waste into the river.

Further downstream alongside Cajamarca city, a dam in the Mashcon river (Figure 3c) induced algal growth, leading to the highest DO and nitrate peaks (Figure 2). The excessive COD at the last sites (39 and 40, Table A1) confirms the impact of direct sewage discharges. Moreover, the dominance of Chironomidae, Tubificidae and the small taxa diversity yielded a very bad biological quality (Figure 3c). A combination of nauseous smells, surfacing foams and thick-dark sludges was also present.

## 4. Discussion

### 4.1. Water Quality Assessment

Impacts from economic activities on the watershed increased from upstream to downstream, with a clear exacerbation in the city outskirts. Physicochemical measurement deviations (Figures A1 and A2), severe biological impairment (Figure 3c) and multiple measurements exceeding the Peruvian quality standards (Table A1) confirmed the worst deterioration within the catchment. Overall, the water quality pattern consisted of stable values for most measurements in the mine and rural areas and suddenly increasing values of EC, chlorophyll-a, COD, phosphorous and nitrogen compounds in the city (Figure 2). Sampling sites above 3200 m.a.s.l. are less populated and thus less polluted despite their proximity to the mine. Freshwater quality variations at midstream sites are influenced by

rural economic activities and by mixing with other streams. The latter is confirmed by the lowering sulphate concentrations downstream of site 21 in river Grande, exactly after mixing with river Purhuay (Figure 2d, the broader valley between sites 21 and 11). The most populated and widest river stretch is characterized by exacerbating organic, iron and manganese pollution (Table A1) and plummeting oxygen concentrations (Figure 2c, green line). It should be noted that oxygen solubility in high mountain rivers is hindered by the lower oxygen partial pressure, whilst the cold Jalca atmosphere and topographic variabilities increase oxygen solubility and mechanical oxygenation.

Hydromorphological alterations have been reported to pose stronger pressures for benthic communities than physicochemical variations [40]. The riparian vegetation, which was abundant in the midstream subsystem, contributes to the attenuating agricultural land-use impacts on macroinvertebrates [41]. Presumably, the poor biological quality in the river Porcón (top-left corner of Figure 3c, site 5) is mostly influenced by stone extraction from this river. Such economic activity is widespread in the area and has severe impacts on the composition of riverbanks and river substrates. In fact, anecdotes in Cajamarca suggested that freshwater quality deterioration would be worse in the mine-recharged river. Conversely, mining impacts were shown to be less severe than those from other economic activities. However, conventional WQM approaches hardly capture complex, long-term and indirect mining pressures in comparison to hydromorphological and high pollutant concentration (i.e., organic, metals or other toxicants) pressures [42].

As shown in the supplementary KML file (see online version), a reforested mining waste center known as San Jose [43] is near the headwaters of streams corresponding to the highest sampling points (top-right corner in Figure 3a). Since these high-altitude brooks have a near-pristine hydromorphology, the moderate biological quality at site 23 and the iron peak at site 25 could be the result of leachates or groundwater inflows altering microhabitat quality [44], besides the fact that heavy rain and flooding events are able to mobilize contaminated mineral grains by eroding and resuspending mining tailings into fluvial systems [45]. Nonetheless, two acidic brooks in the mine subsystem are likely of natural origin, since their topographical situation is unrelatable to mining drainage or legacies (sites 30 and 34). A possible explanation for such low pH values is that hydrogeological processes causing acidity of headwaters have been described before in Cajamarca [46].

*4.2. Challenges for Water Quality Monitoring of Mining Watersheds*

Based on the Mashcon case study, we denoted challenges for more integrated WQM in mining contexts related to the following:

- The selection of environmental quality references conceived in different contexts than the studied one, including freshwater quality indicators for hallmark mining impacts which were absent.
- The lack of ecological indicators for complex mining impacts, since acid- and metal-tolerant 'good-quality' macroinvertebrates were present, and other potential ones (e.g., Acari, Grypopterygidae, ecotoxicological or microbial enzymatic activity) are uncharacterized in the catchment.

In fact, although our results revealed that urban pollution is a major problem in the mining watershed, the contribution from WQM to sustainable mining operations remains limited by the insufficient integration of ecological backgrounds, biophysical processes and land-use patterns in relation to (pre-)mining activities. This level of complexity in mining impact assessment invokes a rather systemic view of the problem. Therefore, the challenges revealed by the present study are described next at the scientific level as well as for systemic aspects.

4.2.1. Ecological Knowledge Challenges

Naturally acidic waters in Jalca ecosystems provide the necessary microhabitat conditions for the long-term adaptation of benthic bioindicators to acidic waters. Moreover, macroinvertebrate taxa found in the mine subsystem are known to tolerate mining pollution. Benthic bioindicators from the Coleoptera order resist acid–metal pollution [47]. Likewise, bioassessments outside of Peru have

demonstrated that insects from the Plecoptera and Trichoptera orders (which indicate good quality in saprobity-based indices) are tolerant to mining–industrial pollution [48] and low metal pollution [49]. Furthermore, the distribution of Acari macroinvertebrates confirms the need to investigate their potential as ecological indicators in mountainous and mining watersheds [50]. Therefore, mining impacts on freshwater habitats are hardly captured due to the incomplete characterization of site-specific bioindicators, ecological networks and abiotic conditions (e.g., hydrological, geochemical, physical composition or physicochemical patterns) [45] of the Mashcon catchment. Overall, the lack of an environmental baseline hinders the discernment between mining impacts and natural causes of ecological gradients

The integration of ecological knowledge in freshwater quality components, practices and regulations is not straightforward [8]. Contrary to the river continuum concept [51], high-altitude headwaters receive low organic inputs and shading from relatively scarce riparian vegetation. In fact, the development of the river continuum concept was largely based on studies in temperate regions and might not be fully applicable to regions with different physiographic, climatic and ecological patterns [52]. Restriction of vegetation occurs naturally due to abiotic conditions of the high Andes (e.g., oxygen vapour pressure, temperature or precipitation) [53], while the absence of riparian vegetation at lower altitudes is due to increasing anthropogenic pressures. The comprehensive characterization of altitudinal variations in floral assemblages and hydromorphology is needed to differentiate between reference and non-reference monitoring sites in Andean rivers [54].

### 4.2.2. Data Acquisition and Processing Challenges

Long-term consequences of mining on freshwater communities require bioassessments referring to pristine ecosystem conditions. Substantial biotic and abiotic records representing ecological networks, environmental flows, and baseline concentrations of water quality components are cornerstones for assessing unconventional mining impacts on ecological quality, such as the impacts of low metal concentrations [55] or flow magnitude alterations [27].

In addition, chemical speciation, compartmentalization, bioaccumulation or toxicological effects of metals are not enforced in WQM practices. The inclusion of metal bioavailability as a freshwater quality component has been recommended in mining-impacted rivers as a more precise indication of mining pollution, preventing the underestimation of metal concentrations in pH-buffered rivers [56]. Moreover, besides the fact that pH and microbial enzymes influence metal solubility in freshwaters [57], the site-specific physicochemical environment (e.g., the presence of dissolved organic carbon or cationic stress) enhances the ecotoxicity of low zinc concentrations [55]. Metal phase distribution is also affected by pH variations, inducing metal precipitation or affinity to a matrix [56].

Physicochemical variability in the high-altitude tributaries is likely to be natural since these tributaries are at different mountain reaches and have varied hydromorphological characteristics. The stable values in the main watercourse, on the contrary, are more likely to be maintained by MYSRL [58] in the dry season. Data-driven water quality indicators are subject to information loss. For instance, a deviation from neutrality remained "excellent" according to WATQI, whereas Prati determined a lower quality (Table A1). Likewise, although Spearman's rank correlation coefficients suggested that DO is correlated with ammonia and nitrite more than with nitrate (Figure A3), nitrate peaks in urban areas were clearly enhanced by oxygen exacerbation, which in turns favors nitrification. Conversely, the decrease of nitrate and the suddenly increasing nitrite in the city (Figure 2f) are explained by DO being at its lowest levels (i.e., creating anoxic conditions for denitrification), besides the fact that organic pollution stimulates denitrifying bacteria. In addition, the ammonium peak at urban areas (Figure 2e) is likely to be enhanced by the ammonification of organic matter besides direct inputs.

Furthermore, site 25 showed a metal peak and acidic pH, although with good biological quality, whereas in site 23, we found a neutral pH, no metal peaks, but a deteriorated biological quality. In fact, these two sampling sites are downstream of a mining waste center known as San Jose and deserve

major attention due to the potential catastrophic failure from the collapse of embankments containing mine by-products despite being reforested.

### 4.2.3. Systemic Challenges

As a means of mining's social licensing, WQM programs using physicochemical and biological indicators have flourished in Peru after the mediation of the Compliance Advisor Ombudsman due to social–environmental conflicts [11]. Currently, around 44 committees of participatory WQM exist in 11 mining regions of Peru. However, with these practices being restricted to certain locations [59], disparate in points of view [60] and having low academic involvement [61], the WQM soundness debate remains open.

At the regulatory level, Peruvian laws allow pH values between 6 and 9 for any type of water use. Thus, the mining company uses quicklime to neutralise acidic waters, besides adding flocculants and coagulants to separate metals and other particles [30]. However, environmental policies around mining must integrate demographic changes and human water security together with freshwater biodiversity protection measures [62]. For instance, efforts by MYSRL to support slope farmers of Cajamarca with sprinkler irrigation technology were shown to have attenuative effects on agricultural run-offs and soil erosion [63], and such experiences must proceed to the decision-making arena.

Furthermore, the typology of Peruvian freshwater streams, currently based on topology and priority uses (i.e., anthropogenic functions) [64], needs to account for ecosystem variability and functionality, as reflected in the 15 terrestrial ecoregions proposed for the Red Book of Endemic Plants of Peru [13]. An example of biophysical knowledge integration in WQM regulations is the German typology of 25 running waters throughout the entire country, since each type of stream has its own ecological, hydromorphological and geological characteristics; thus, there is a requirement for type-specific references and assessment procedures for freshwater quality components [65]. Regarding bio-indicative potential, Canadian and Australian WQM protocols have included Acari in biotic scores thanks to local studies attaining higher taxonomical resolution [50].

Updating environmental quality references and assessment protocols implies encompassing social–ecological system dynamics. For instance, with water scarcity being a major concern for non-mining stakeholders, MYSRL dams were built for storing water and retaining sediments [15]. However, such stream impoundments interrupt aquatic migration and alter freshwater abiotic conditions such as temperature, flow velocity, and dissolved oxygen concentration [66], which in combination with urban pollution trigger severe pollutant dynamics, as shown by this study. In fact, a combination of anaerobic, anoxic and aerobic conditions in water bodies has ecological implications in the functioning of biodegradative processes and mutualistic microbial relationships affecting freshwater's self-purification capacity [67].

Furthermore, the degree of mining's impacts on aquatic habitats and freshwater ecosystem services vary at each lifecycle stage of mining, raising multiple sustainability concerns [68]. While we confirmed that physicochemical quality is preserved in the mining-recharged headwaters, questions arise regarding how much of the anthropogenic impact on the Mashcon watershed is attributable to mining development. Will the headwaters' quality be maintained once the mine is closed? How will the multiple mining legacies (e.g., reforestation, groundwater rebound effect, agricultural enhancements and urban growth) affect freshwater quality? How will non-mining stakeholders be affected in post-mining scenarios? Natural flow regime alterations will affect the ecosystem state. The potential release of chemicals and acid drainage formation during and after mine exploitation have a negative effect on soil quality (acidification, accumulation of toxic elements) and thus on water quality [69]. The human migration linked with mining induces a more intensive use of local resources, which induces land cover changes and water quality deterioration if waste water is not treated. The removal of vegetation induces soil erosion and soil degradation. A reduced soil quality in turn is also adverse for vegetation. The overall reduction of the ecosystem state causes a decline in ecosystem service

production, such as the impaired mediation of mass and liquid flows or lower availability of plants and animals for non-mining stakeholders [68].

In the common understanding of mining sustainability researchers [5,11,70,71], the above-mentioned concerns are not solely a matter of applied science but are primarily dependent on a combination of decision-making, technological and operational viabilities. In fact, WQM implementation is subject to available budgets, technical capabilities and political structures, besides natural constraints such as climate, topography and ecological understanding (specific to each social–ecological system) [72]. The scarcely-studied Andes region is characterized by its topographic and climatic complexity [18], making ecosystem measurements prone to variability. Regarding social spheres, highlands such as the Andes are characterized by socio-economic inequalities and political exclusion of less-developed populations [19]. The decentralization process which has been taking place in Peru for more than a decade has led to the distribution of mining fiscal revenues among producing and non-producing districts, aiming to boost human well-being and compensate for any negative mining impact [12]. However, the great deal of attention to socio-economic priorities in mining contexts often neglects the strong dependence of human well-being on ecosystems' health and functions [68].

### 4.3. Windows of Opportunity

Based on our discussion, Figure 4 depicts hindrances (red lines) of WQM for the protection of freshwater ecosystems. Corrective measures are also depicted (blue lines), suggesting the allocation of mining revenues for the WQM developments and integration of WQM practitioners for the prevention of unsustainable mining watershed exploitation. Integrating a scientific basis in environmental monitoring is feasible via collaborations among governments, academia and businesses. For example, guidelines for freshwater bioassessments are available for Peru thanks to a joint effort between the Natural History Museum, the Major National University of San Marcos and the Ministry of Environment [73]. Likewise, environmental monitoring developments in the context of natural resource exploitation and problematic stakeholder relationships have taken place in Peru before. Long-term baseline records of six biological components were obtained for purposes of biodiversity protection and the adaptive management of the Camisea gas project in the Peruvian Amazon. Such a multidisciplinary approach came from a joint venture between the Smithsonian Institute and Shell Prospecting and Development Peru and included the participation of local communities and public bodies. Regrettably, lobby disagreements impair such efforts [74]. Likewise, embedding the iterative adaptation (i.e., learning) of environmental monitoring components has been proposed to address multi-stakeholder disagreements around eco-tourism in Peru. Such an adaptive monitoring solution [75] has been proposed in response to similar social-ecological issues to those in our case study [15].

Governments should enhance WQM information systems. Understanding that current WQM programs in Peru target different water quality components (e.g., one institution measuring coliforms while another focuses on chemical pollution, and a different one manages weather stations), the National System for Environmental Information (SINIA) is a promising prospect for WQM integration. Eventually, interinstitutional cooperation will also be needed for the intercalibration and standardization of WQM, including hydrological process integration with water quality dynamics and a multi-jurisdiction perspective of river basins [7,56] (e.g., the Mashcon is part of the multi-national Amazon river basin).

Advancing WQM also pertains to mining business competitiveness, which should align to a green economy and sustainability paradigms [76]. In fact, the interest of mining companies in protecting ecosystem integrity is synergized by the need for green economy competitiveness and the growing public awareness and subsequent claims of environmental responsibility. Therefore, proactive measures in the different spheres influencing the scientific, political and social soundness of WQM (Figure 4) are essential for sustaining mining businesses and human well-being in the long term.

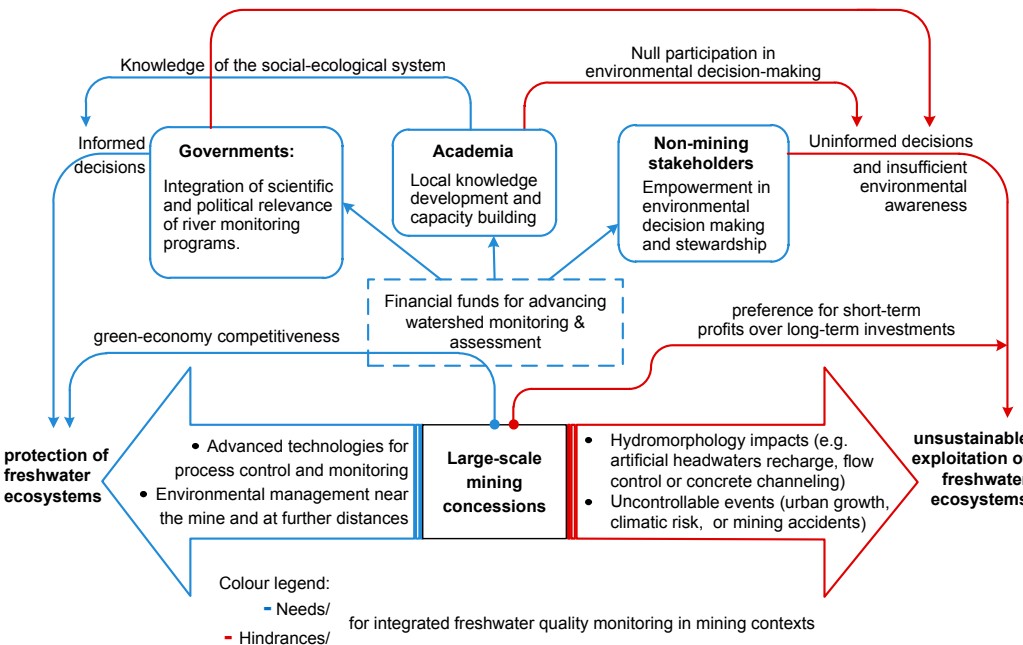

**Figure 4.** Systemic recommendations for advancing the integrated freshwater quality monitoring of mining watersheds. The red color indicates the hindrances ultimately leading to unsustainable freshwater resources usage. The blue color indicates the need to integrate freshwater quality assessments, bringing stakeholders together to prevent unsustainable watershed exploitation.

## 5. Conclusions

We found that direct mining impacts could not be detected by the selected freshwater quality parameters, in contrast to the severe impacts from urban pollution. However, a quantitative determination of freshwater quality components in mining contexts might be a weak assessment endpoint if (social–ecological) system knowledge is insufficient for mining watersheds. Ecological knowledge gaps remain due to limitations of database availability and high-mountain expert knowledge and the lack of multidisciplinary integration. Determining the long-term effects of controlled mining activities requires ecotoxicological, microbial and groundwater process knowledge to capture ecological losses. Understanding the consequences of ecological losses rather than relying solely on data-driven quality thresholds is crucial as mining causes complex changes in the overall system. The outlook from our assessment suggests that a system-based understanding is urgently required to expand the traditional view of mining's impacts on freshwater ecosystems, integrating governments, scientific disciplines and mining businesses in the protection of aquatic habitats and non-mining stakeholders.

**Supplementary Materials:** The following are available online at http://www.mdpi.com/2073-4441/11/9/1797/s1, Supplementary material S1: Extended case study information, selection and pictures of the sampling locations and summary of laboratory analyses, and Supplementary KML file: hydrographic map with sampling locations.

**Author Contributions:** Conceptualization, D.M.-G., G.W. and P.G.; methodology, D.M.G., E.B., J.V.B. and N.D.S.; software, E.B. and C.V.B.; investigation, D.M.G., E.B., J.V.B., M.S.P. and M.A.E.F.; resources, N.D.A., K.A.C.D.S., G.W. and P.G.; data curation, D.M.G., E.B., J.V.B., C.V.B., N.D.S. and K.A.C.D.S.; writing—original draft preparation, D.M.G., E.B., M.S.P., M.A.E.F. and P.G.; writing—review and editing, D.M.-G., M.S.P., K.A.C.D.S, G.W. and P.G.; visualization, D.M.-G., E.B. and C.V.B.; supervision, G.W. and P.G.; project administration, D.M.-G., N.D.A., G.W. and P.G.; funding acquisition, D.M.-G., N.D.A., G.W. and P.G.

**Funding:** This research was supported by FONDECYT-CONCYTEC [grant contract number 002-2016-FONDECYT], and by a VLIR-TEAM project [ZEIN2013PR395: 'Impact on surface water resources and aquatic biodiversity by opencast mining activities in Cajamarca, Peru']. The APC was funded by the Aquatic Ecology (AECO) Research Unit.

**Acknowledgments:** The authors thank Koen Lock for the taxonomic identification of macroinvertebrates, as well as to the members of the National University of Cajamarca who contributed to the VLIR-TEAM project.

**Conflicts of Interest:** The authors declare no conflict of interest.

## Appendix A

**Table A1.** Composite Indices of physicochemical quality (Prati and WATQI) and Peruvian water quality standards that were not met for freshwater streams intended for potable water production. For dissolved oxygen (DO), standards are minimum permissible values. For pH, acceptable ranges are provided. Standards for turbidity, dissolved metals (Fe and Mn), total phosphorus (Total P), ammonia (NH4-N) and chemical oxygen demand (COD) are maximum permissible values.

| Sampling Site ID | PRATI | Class | WATQI (%) | Class | Peruvian Water Quality Standards: | DO (†6 mg/L) (‡4 mg/L) | Turbidity (†100 NTU) (‡5 NTU) | Total P (†0.15 mg/L) (‡0.1 mg/L) | NH4-N (†N.A. mg/L) (‡1.5) | COD (†30 mg/L) (‡10 mg/L) | pH (†5.5-9.0) (‡6.5-8.5) | Fe (†5 mg/L) (‡0.3 mg/L) | Mn (†0.5 mg/L) (‡0.4 mg/L) |
|---|---|---|---|---|---|---|---|---|---|---|---|---|---|
| 1 | 1.554 | Acceptable | 87 | Excellent | | | | 0.25 * | | | | | |
| 2 | 1.015 | Acceptable | 96 | Excellent | | | 27 | | | | | | |
| 4 | 0.536 | Excellent | 91 | Excellent | | | | | | 16 | | | |
| 5 | 0.485 | Excellent | 94 | Excellent | | | 7 | 0.13 | | 15 | | | |
| 6 | 0.478 | Excellent | 86 | Excellent | | | 366 * | 0.61 * | | | | | |
| 7 | 0.517 | Excellent | 94 | Excellent | | | | | | | | | |
| 8 | 0.442 | Excellent | 96 | Excellent | | | | | | | | | |
| 9 | 0.394 | Excellent | 96 | Excellent | | | | | | | | | |
| 10 | 0.427 | Excellent | 96 | Excellent | | | | | | | | | |
| 11 | 0.43 | Excellent | 96 | Excellent | | | | | | | | | |
| 12 | 0.435 | Excellent | 96 | Excellent | | | | | | | | | |
| 13 | 0.417 | Excellent | 96 | Excellent | | | | | | | | | |
| 14 | 0.421 | Excellent | 96 | Excellent | | | | | | | | | |
| 15 | 0.398 | Excellent | 93 | Excellent | | | | | | | | | |
| 16 | 0.575 | Excellent | 94 | Excellent | | | | | | | | | |
| 17 | 0.515 | Excellent | 96 | Excellent | | | | | | | | | |
| 18 | 0.517 | Excellent | 96 | Excellent | | | | | | | | | |
| 19 | 0.439 | Excellent | 95 | Excellent | | | 11 | | | | | | |
| 20 | 0.48 | Excellent | 95 | Excellent | | | 11 | | | | | | |
| 21 | 0.613 | Excellent | 94 | Excellent | | | | | | | | | |
| 22.1 | 2.272 | Slightly polluted | 64 | Good, pure | | | | | | | | | |
| 22.2 | 1.054 | Acceptable | 86 | Excellent | | | | | | | | | |
| 23 | 0.387 | Excellent | 95 | Excellent | | | | | | | | | |
| 24 | 2.235 | Slightly polluted | 64 | Good, pure | | | | | | | | | |
| 25 | 0.717 | Excellent | 91 | Excellent | | | | | | | 6.3 | 0.39 | |
| 26 | 0.816 | Excellent | 93 | Excellent | | | | | | | | | |

**Table A1.** *Cont.*

| Sampling Site ID | PRATI | Class | WATQI (%) | Class | Peruvian Water Quality Standards: | DO (†6 mg/L) (‡4 mg/L) | Turbidity (†100 NTU) (‡5 NTU) | Total P (†0.15 mg/L) (‡0.1 mg/L) | NH4-N (†N.A. mg/L) (‡1.5) | COD (†30 mg/L) (‡10 mg/L) | pH (†5.5-9.0) (‡6.5-8.5) | Fe (†5 mg/L) (‡0.3 mg/L) | Mn (†0.5 mg/L) (‡0.4 mg/L) |
|---|---|---|---|---|---|---|---|---|---|---|---|---|---|
| 27 | 0.767 | Excellent | 93 | Excellent | | | | | | | | | |
| 28 | 0.851 | Excellent | 94 | Excellent | | | | | | | | | |
| 29 | 0.73 | Excellent | 94 | Excellent | | | | | | | | | |
| 30 | 0.828 | Excellent | 85 | Excellent | | | | | | | 5.7 | | |
| 31 | 0.56 | Excellent | 94 | Excellent | | | | | | | | | |
| 32 | 0.579 | Excellent | 94 | Excellent | | | | | | | | | |
| 33 | 0.633 | Excellent | 94 | Excellent | | | | | | | | | |
| 34 | 1.354 | Acceptable | 74 | Good | | | | | | | 4.6 * | | |
| 35 | 0.497 | Excellent | 96 | Excellent | | | | | | | | | |
| 36 | 0.469 | Excellent | 95 | Excellent | | | | | | | | | |
| 37 | 0.731 | Excellent | 94 | Excellent | | | 27 | | | | | | |
| 38 | 0.74 | Excellent | 94 | Excellent | | | 37 | 0.12 | | | | | |
| 39 | 5.337 | Polluted | 44 | Moderate | | 2.8 * | 12 | 1.24 * | 7.45 * | 31 * | | | 0.64 |
| 40 | 6.564 | Polluted | 20 | Heavily polluted | | 1.7 * | | 2.30 * | 9.21 | 47 * | | 0.34 | 0.47 |

† Peruvian water quality standard for streams requiring advanced treatment. ‡ Peruvian water quality standard for streams requiring disinfection. * The value exceeds both water quality standards.

According to the WATQI index, 35 of the 40 sites had "excellent" water quality, whilst Prati indicated "excellent" quality only in 32 sites. This discrepancy consisted in having "acceptable" quality according to WATQI at sites where Prati indicated "excellent" quality (sites 1, 2 and 22.2). This WATQI–Prati discrepancy also occurred as "slightly polluted" instead of "good" quality (sites 22.1 and 24), and as "polluted" instead of "heavily polluted" (site 40). In general, both indices confirmed the longitudinal water quality pattern: both in the river Porcón and in river Grande, all sites had "excellent" physicochemical quality, whilst downstream water quality decreased to "acceptable" at the urban core, and then to "heavily polluted" at the city outskirts.

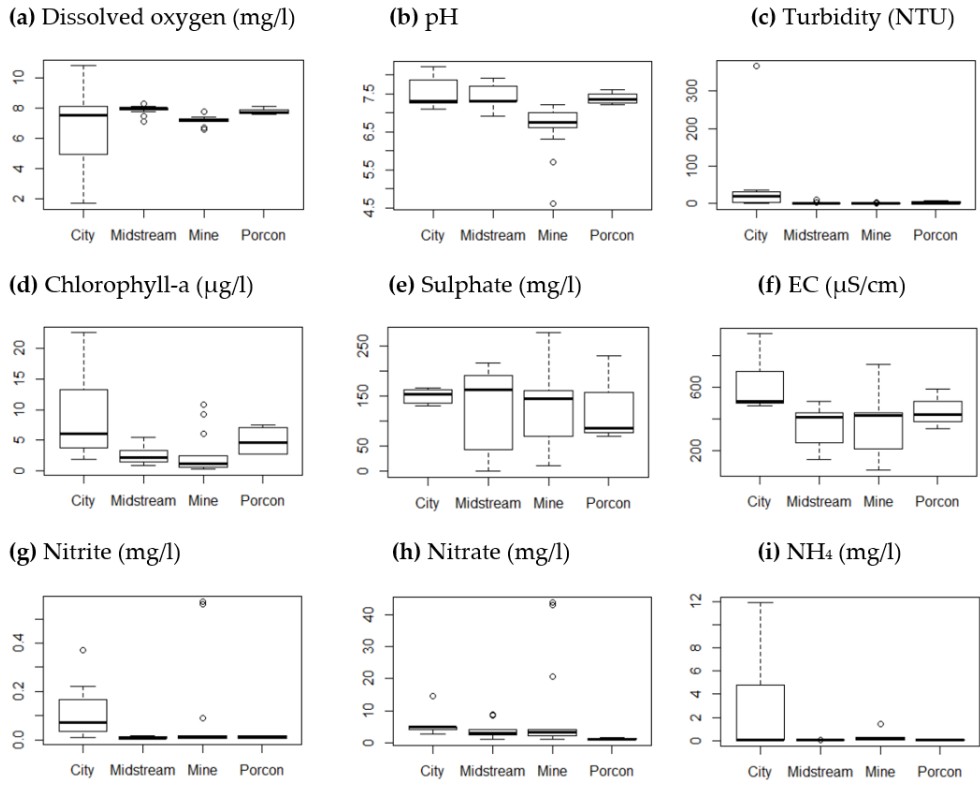

**Figure A1.** Boxplots of physicochemical measurements pooled by sampling subsystems (see Table 1). The boxplots show that pH, sulphate and electric conductivity (EC) distributions have differentiating features, whereas the rest of the parameters show no clear differences among boxplots. Alkaline outliers (i.e. pH≈8) belonged to the most polluted river stretches, whereas the alkalinity of sites 17, 18, 19 and 20 occurred together with low EC, nitrogen and phosphorus (i.e., unpolluted waters). In the tributary Río Porcón, an outlying SO4- concentration matched concentrations found at much higher altitudes, such as sites 21, 22.1 and 24.

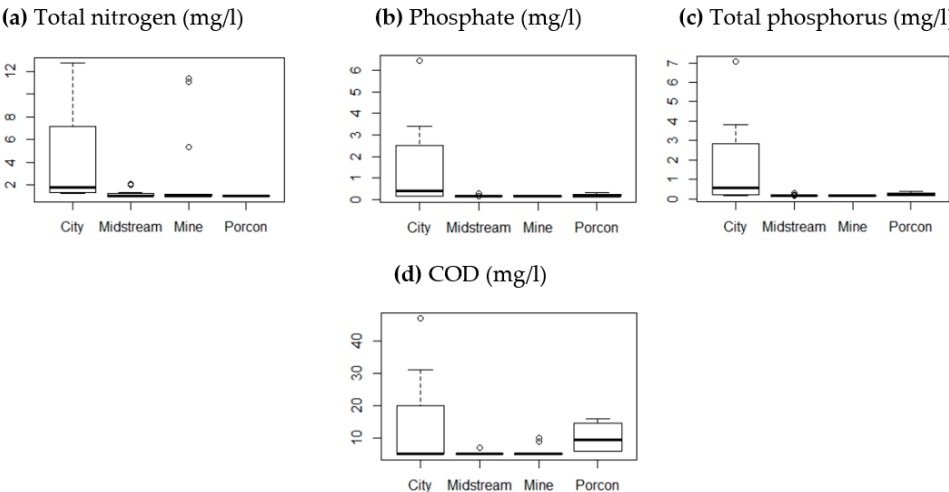

**Figure A2.** Boxplots for total nitrogen, phosphates, total phosphorus and chemical oxygen demand (COD). Furthest outliers belong to the most-downstream sampling sites in the city (heavily polluted). Boxplots helped in comparing the data distribution among different data subsets, including the visualization of outliers and variability in each group. Several outliers confirmed the anthropogenic alteration of water quality, as well as the effects of an irrigation outflow in the mine subsystem.

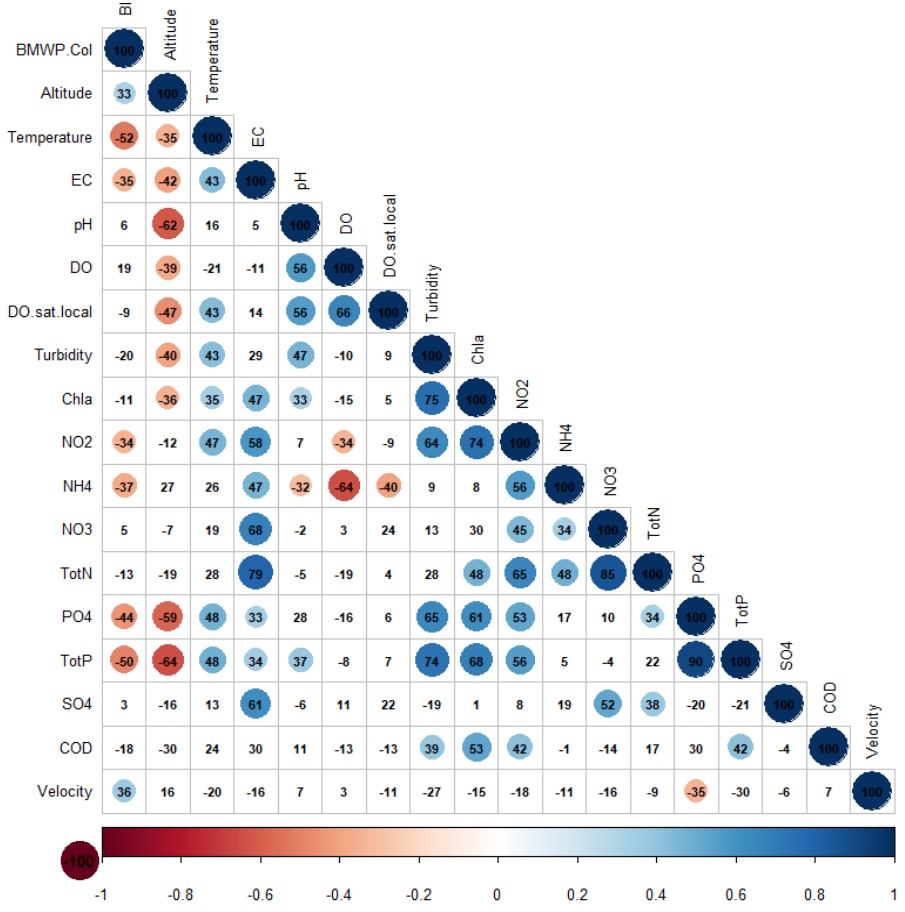

**Figure A3.** Spearman's rank correlation results. Significant correlations ($p < 0.05$) are indicated with a circle, and the 'rho' ($\varrho$) coefficients (which range from -1 to +1, as indicated in the colour legend) are multiplied by 100 inside each cell. The circles' radius and color indicate the strength and direction of the correlation, respectively. A lower-limit circle color is depicted next to the color legend, and the corresponding upper-limit circle color is at the top of each column.

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
