# Peer review of "Assessing the Freshwater Quality of a Large-Scale Mining Watershed: The Need for Integrated Approaches"

_water, doi:10.3390/w11091797_

Round 1

Reviewer 1 Report

Review of Water-556527, Mercado et al, 2019

This paper describes in detail a catchment-scale study on the effects of multiple pressures (including mining) on the physico-chemical and biological water quality of the rivers. As no clear effects of mining came out (partly due to limitations in the methodology and indicators used), the paper broadens to a discussion on socio-ecological aspects and sustainability of mining activities, in catchment context, in general.

Both topics certainly deserve publication in the international literature. But I feel that this paper stands on ‘two legs’, and that these ‘legs’ should be distinguished better, either by a clearer separation between sections, or, maybe better, by splitting the manuscript into two papers. One on the WQ study, describing the results and discussing the ins and outs of the different indicators (what is wrong with them?) and recommendations for improvement. And a second one on the feasibility of sustainable catchment management including mining. Where you can also build on the results of other studies without having to excuse yourself all the time that the current study did not reveal clear impacts.

Reflecting these two ‘legs’, the Introduction is poorly structured. It mixes a lot of issues which are all important in themselves (methodological aspects of WQM, indicators, multidisciplinarity, social aspects, catchment approach, etc.), but the focus is not clear. The Introduction should end with a clear description of the aim and research questions of the study.

The same applies to the Discussion: the section on socio-ecological aspects of mining activities is of course highly relevant in general, but does hardly relate to the current WQ study.

Furthermore, I recommend a clearer distinction between Results and Discussion, also within the parts on the WQ study itself. i.e. keep the Results section more neutral, and much of the text of 3.2, 3.2 and 3.3 can be moved to the Discussion.

Other comments:

Abstract:

L21: regard -> study

L26: few: a few

L27 (and elsewhere in the paper:) “livelihood activities”: I’d prefer ‘(human) pressures’, or ‘economic activities’

L25-27: I suppose the point here is that mining effects do not pop up from the indicators used, in contrast to (less important) effects of agriculture. I suggest to state that clearer.

Introduction:

L96: what do you mean by ‘barriers at the normative level’?

Methods:

L108-9: what fraction of the river water is extracted (about)?

L111-2: who is abstracting the groundwater, the mining industry? And what is the source of the recharge, is that discharge from the mines?

L113-4: give a brief indication of what kind of pollution.

L121: Mention that ‘Porcon’ is a tributary (subcatchment) of the Rio Grande (or Mashcon).

L156-7: Briefly explain what is comprised in the Prati and WATQI indices.

Results:

Section numbering: 3.2 appears twice.

L253-4: Between the lines it seems that the mining industry is obliged to keep the pH of the river neutral, is that correct? If so, state this more clearly.

L274-6: This sentence is unclear. “evince”: show?

Figure numbering: A2 appears twice.

Table 2: the presentation of only the sites that exceed the standards gives the impression of a biased paper (which it is not). Combine this info with Table A1 in the Appendix.

Figures A1 and the first A2: is this the most informative way of clustering the sites, in view of the differences between upper, middle and lower reaches of the main stream shown earlier?

Figure A2 (second): add that the numbers are rho x 100. Did you use a lower limit for the depiction of a coloured circle?

Discussion:

L326-334: unclear what is the message here. You mean that the indicators of your study could not distinguish the effects of the different pressures? Please clarify.

L330-31: I understood earlier that some of the upper sites were not impacted by the mines, is that not correct?

L341: “uncharacterized”: ?

L342: that is a very broad statement. Skip, and keep to the concrete issues.

L343-9: what is the point you want to make here?

L351: contrary to the RCC: explain a bit further why you consider the RCC not valid for Andean streams.

L357, 361: typification: typology.

L384: dependent

Figure 4, top line (“Overarching issue”): it is not very clear what you mean exactly.

L409-10: which company?

Author Response

Dear Reviewer 1,

We would like to extend our gratitude for the dedicated reading and constructive remarks to our manuscripts. We have revised our manuscript accordingly. We acknowledge that the recommended modifications improve the quality of our manuscript. We hope that the changes and explanations are acceptable and satisfactory.

You can find in attachments the details of the modifications and explanations point by point.

Yours sincerely,

Daniel Mercado García

Department of Animal Sciences and Aquatic Ecology

Ghent University, Belgium

E-mail address: [email protected].

Reviewer 2 Report

(1)The authors should explain the choice of sampling in the dry season. Reasons could include  best season for detection of problems and connection of water quality to land use, common practice 

References that could be used: 

Mackay A.K. et al.   (2011)    Water and sediment quality of dry season pools in a dryland river system: the Upper Leichhardt River, Queensland, Australia.  J. Envir. Monit. 13:  2050-2061. 

Ackerman D. et al.  (2005)  Dry-season water quality in the San Gabriel River watershed.  Bull. Southern California Acad. Sci. 104(3)  125-145.

Quagraine  E.K., Adokoh C.K.   (2010)  Assessment od dry season surface, ground, and treated water quality in the Cape Coast municipality of Ghana.  Envir. Monit. Assess. 160: 521-539

(2)The authors mention possibly continuing problems after mining ceases but do not emphasize how large a problem this in.  Another problem that should be emphasized is detection of sites capable of producing catastrophic failure from collapse of dams/embankments containing mine byproducts.

References that could be included:

Jones J.I. et al. (2018) The Impact of Metal-Rich Sediments Derived from Mining on Freshwater Stream Life. In: de Voogt P. (eds) Reviews of Environmental Contamination and Toxicology Volume 248. Reviews of Environmental Contamination and Toxicology (Continuation of Residue Reviews), vol 248. Springer,

Park I et al., (2019)  A review of recent strategies for acid mine drainage prevention and mine tailings recycling.    Chemosphere 219: 588-606.

Author Response

Dear Reviewer 2,

We are thankful for your dedicated reading time as well as for providing constructive remarks and references for our research. We have revised our manuscript accordingly. We acknowledge that the recommended modifications improve the quality of our manuscript. We hope that the changes and explanations are acceptable and satisfactory with the expectation of the editors and reviewers.

You can find in attachments the modifications and explanations point by point.

Yours sincerely,

Daniel Mercado García

Department of Animal Sciences and Aquatic Ecology

Ghent University, Belgium

E-mail address: [email protected].

Round 2

Reviewer 1 Report

To my opinion, the authors have adequately replied to the comments and have greatly improved the structure of the paper. I would recommend the paper now for publication.

May be two small errors:

L26: I think deterioration is a better word.

L267: I suppose 32000 m should be 3200 m.